# Car/Motorbike Drivers’ Willingness to Use and to Pay for Alcohol Interlock in Taiwan

**DOI:** 10.3390/ijerph182111516

**Published:** 2021-11-02

**Authors:** Rong-Chang Jou, Yi-Hao Lu

**Affiliations:** Department of Civil Engineering, National Chi Nan University, Nantou 545301, Taiwan; s45678931@gmail.com

**Keywords:** drunk drivers, alcohol interlock, exploratory factor analysis, alcohol use disorders identification test, double-hurdle

## Abstract

This study explored the important factors affecting drunk car/motorbike drivers’ willingness to use and pay for alcohol interlocks. Data were obtained through a survey upon choice-based sampling conducted in central Taiwan. Questionnaires were distributed to the participants of drunk driving and road safety education courses from 17 August to 26 October 2020. All drunk drivers whose driver’s licenses are revoked for drunk driving are mandated to participate in this course. Prior to the survey, the researchers explained the questionnaires, instructed the participants to complete the questionnaires, and then collected all the questionnaires. The socioeconomic characteristics of drunk drivers, awareness of alcohol interlocks and drunk driving, drinking patterns and health self-assessment before and after drunk driving ban enforcement, and changes in the number of trips were investigated. This study applied the double-hurdle model for data analysis to estimate the variables affecting drunk car/motorbike drivers. Results indicate that the respondents who were classified by the Alcohol Use Disorders Identification Test as high-risk drinkers before and after drunk driving ban enforcement were more willing to use alcohol interlocks and to pay higher prices. Additionally, the respondents with declined health self-assessments were also more willing to use alcohol interlocks and pay higher prices. This study suggests offering subsidies for alcohol interlocks to families with financial difficulties, in order to increase the alcohol interlock installation rate. Moreover, since the current duration of license suspension and withdrawal is considerably long, drunk drivers avoid using and installing alcohol interlocks by reducing the number of trips. In other words, the willingness to install alcohol interlocks may be increased by reducing the duration of license suspension and withdrawal.

## 1. Introduction

According to the statistics of the National Police Agency, among A1 (A1 class means someone is killed instantly or dies within 24 h of when the accident occurred regardless of hospitalization or not) class traffic accidents in Taiwan in the past 10 years, the top three causes for accidents caused by drivers’ negligence are related to drunk driving. While the proportion of drunk driving accidents has declined in recent years, drunk driving continues to be a serious issue. As the main cause of road traffic accidents globally [1,2], drunk driving increases the drunk drivers’ risk of injury or death and puts other road users at risk. In order to prevent drunk driving accidents, government agencies have strengthened various prevention measures in terms of policies, legal institutions, promotion, and enforcement, and revised the regulations and penalties for drunk driving more strictly over the years.

The Ministry of Transportation and Communications, Taiwan, implemented a new policy on drunk driving in March 2020, stipulating that anyone with a breath alcohol concentration (BAC) over 0.15 mg/L or a blood alcohol concentration over 0.03% is considered a drunk driver. Fines vary according to the vehicle, ranging from NTD (1 USD = 30 NTD) 30,000 to NTD 120,000 for cars, NTD 30,000 to NTD 90,000 for motorbikes. Those who drive drunk more than three times shall receive treatment for alcohol addiction before taking the test for a new license for cars or motorbikes. Also, the supervisory authority will issue one-year restricted driver’s licenses, and the drivers with restricted driver’s licenses can only use vehicles equipped with alcohol interlocks. Drivers also need to register their main controlled vehicles with the highway supervisory authority. Restricted drivers shall have their equipment inspected, and are required to download their event log data at suppliers’ service centers each month for inspection by the supervisory authority [3]. In Taiwan, while a large number of applicants participated in the driver’s license test from March 2020 to December 2020 after their driver’s licenses were suspended, only a few have installed qualified alcohol interlocks and registered with the supervisory authority, which indicates the poor effect of the alcohol interlock policy.

Many countries have used alcohol interlocks as drunk driving preventive strategies. Ref. [4] studied public attitudes towards alcohol interlocks in Australian samples and measured drinking patterns with the AUDIT (Alcohol Use Disorders Identification Test), and found the following: male and young respondents are more likely to agree that alcohol interlocks are binding on individuals; low-risk drinkers are the least likely to approve alcohol interlock interventions for individuals; high-risk drinkers strongly support that alcohol interlocks are binding on individuals; as the drinking risk level increases, those who have had their licenses suspended due to drunk driving are more likely to use mandatory interlocks rather than counseling alcohol interlocks. Ref. [5] studied first-time drunk driving offenders in Ontario, Canada from 2005 to 2014. The government used incentive measures to increase the alcohol interlock installation rate and addressed the social costs caused by delayed installation. The results showed a 54% increase in the installation rate, a 49% increase in the number of convicted drunk drivers per month, and a sharp reduction in the usual length of a 146-day court trial. However, the recidivism rates of participants in this program at 90 and 180 days increased, indicating that short-term use of alcohol interlocks cannot eliminate drunk driving.

Ref. [6] found that the drunk driving recidivism rate dropped to 1.8% after the implementation of alcohol interlock programs. Based on data from the U.S. alcohol interlock program, [7] found that alcohol interlocks can reduce recidivism, as well as the risk of death caused by drunk driving. Based on the data from Canada’s alcohol interlock program, [8] found that the drunk driving recidivism rate of drivers who install alcohol interlocks voluntarily is lower than that of drivers who are forced to install alcohol interlocks, and both are lower than that of drivers who install no alcohol interlocks. The recidivism rate in the Netherlands was reduced to 4%. Ref. [9] proved that the recidivism rate will not return to the original level after the alcohol interlock program. Ref. [10] indicated that drunk drivers with a more serious drinking pattern, and who have a higher risk for recidivating, are more likely to recognize the potential benefits of the interlock as a DUIA (Driving under the influence of alcohol) preventive countermeasure. 

Taiwan’s alcohol interlock policy was implemented in 2020. Three manufacturers offer alcohol interlock installation and rental services, namely, ALCOLOCK, which is a Canadian firm with a monthly rental fee of NTD 4000; Draeger Safety Taiwan, which has a sale price of NTD 112,350, and; Measuring Instruments Supplier, which charges a monthly rental fee of NTD 5000. Compared with the purchase price of alcohol interlocks (€2000–4000), as stipulated by other national governments, the alcohol interlock price in Taiwan is rather high. This is one of the reasons for the low alcohol interlock installation rate in Taiwan. Since less than ten drivers in Taiwan have currently installed alcohol interlocks, the aim of this study is to explore drunk drivers’ willingness to install and pay for alcohol interlocks. 

Due to the importance of alcohol interlocks to traffic safety and social stability, this study constructed the willingness-to-pay models for car and motorbike driver groups, respectively, in order to understand the differences between drunk car drivers and drunk motorbike drivers regarding their willingness-to-pay and participation willingness. This study applied the double-hurdle model to examine the socioeconomic characteristics of drunk drivers, their awareness of alcohol interlocks and drunk driving, their drinking patterns and health self-assessment before and after drunk driving ban enforcement, and changes in their number of trips. Their willingness to install and pay for alcohol interlocks, as well as the important factors affecting willingness and prices, were further discussed.

The awareness of alcohol interlocks and drunk driving was investigated according to Aker’s social learning theory [11]. The social learning theory, as proposed by [12], is a concept based on social psychology, which focuses on factors that stimulate or depress behaviors to explain deviant behaviors and criminal activities. Finally, the drinking patterns were studied by the Alcohol Use Disorders Identification Test (AUDIT) scale [13], as recommended by the World Health Organization (WHO). The results reveal the factors that are not conducive to alcohol interlock promotion and provide policy suggestions to improve the current ineffective alcohol interlock policies.

The contributions of this study are: (1) to classify drunk drivers’ awareness of alcohol interlocks and drunk driving through exploratory factor analysis (EFA) according to Aker’s social learning theory; (2) to classify drunk drivers’ drinking patterns by the AUDIT; (3) to apply the double-hurdle model to estimate the willingness to install and pay for alcohol interlocks; and (4) to explore drunk car drivers and drunk motorbike drivers, respectively. The inclusion of motorbike drivers has two reasons: one is the high population of motorbike drivers in Taiwan, and another is the dangerous behavior and vulnerability of motorbike drivers under the influence of alcohol [14,15].

## 2. Literature Review 

### 2.1. Literature on Willingness-to-Pay

Ref. [16] studied Irish household expenditures on petrol and diesel, as well as the important factors affecting their decisions. According to the results, the number of people with jobs in a family, residential area, the number of cars owned by a family, and public transportation consumption are the main factors affecting expenditures on fuel (petrol and diesel). As urban public transport is more developed than rural or suburban public transport, fuel costs are low; households with no cars are more dependent on public transport, so that they spend less money on petrol and diesel. 

Ref. [17] discussed the seat pre-selection and value-added service in behavior of Taiwanese passengers who take low-cost flights to Japan. The results indicated that passengers who had experienced purchasing pre-selected seats were more willing to purchase this additional service and spend a higher amount of money for that service. Ref. [18] estimated choice willingness and willingness-to-pay in different scenarios by the double-hurdle model and Tobit model. As the return distance in the scenarios increased, the respondents were more willing to use designated driving services and pay high prices. Young people (under 30), low-income people (less than 20,000), people who drink every day or drive every time after they drink, and people who take taxis or ask relatives and friends to help them go home are less likely to use designated driving services and only pay low prices regardless of distance.

### 2.2. Introduction to EFA and AUDIT

Ref. [19] explored the important factors leading to the high proportion of young Australians who drive after drinking by using EFA, and found that personality structure is related to an increase in the likelihood of harmful drinking and dangerous driving behaviors. The questionnaire was designed according to Akers’ social learning theory. The statistical analysis showed that 40% of the participants had driven after drinking. According to model estimation results, people who scored high on attitudes to drunk driving and alertness to drunk driving punishments are less likely to become drunk drivers, while people who scored high on drunk driving affected by friends and fun-seeking drunk driving are more likely to become drunk drivers.

Ref. [20] used EFA to identify the most suitable factor structure for drivers in New Zealand and explored the correlations between factor outcomes and crash involvement. Ref. [21] applied EFA to consolidate relative questions in numeric factor quantities, and performed logistic regression on the calculated component scores to investigate the effects of each factor on past crash involvement of car drivers. Ref. [22] used driver’s self-evaluation data to define which elements cause visual and cognitive distraction. EFA, Confirmatory factor analysis (CFA), and Structural Equation Modeling (SEM) were implemented. Results showed that the impacts of different factors on drivers’ perception of crucial changes in the traffic environment varied.

Drivers who drink often suffer from alcohol use disorder. AUDIT is used for assessments of alcohol misuse [23,24].

AUDIT consists of two parts. The first part asks about the respondents’ daily drinking habits, including amount, drinking frequency, and times of excessive drinking; the second part is about the effects of drinking on respondents’ daily lives, including sleep quality, alcohol dependence, sense of guilt, frequency of alcoholic abstinence advice from relatives, and diagnosis of alcohol addiction. Each item is self-rated on a scale of 0 to 4, as shown in [23,24]. There are four stages according to the total score of the two parts: high risk (>19), medium risk (16–19), low risk (8–15), and very low risk (1–7). It is recommended to change the alcohol consumption depending on the risk, and the diagnosis of alcohol addiction is conducted for those at high risk. The AUDIT scale is shown in Table 1 [13].

## 3. The Model

This study aimed to discuss drunk car/motorbike drivers’ willingness to use and pay for alcohol interlocks. This study applied the survey method and econometric model to collect the data and estimate the related significant influencing factors and their effects. This study asked drunk drivers to give the maximum amount that they are willing to pay through open-ended bidding. The double-hurdle model was employed to measure the effects of different variables on the participants’ decisions to use and pay for alcohol interlocks in two phases. This study adopted the Limdep statistical software to calibrate the coefficient of the model. The following account further describes the principles of the model.

### 3.1. Survey Method and Statistical Analysis

This study adopted choice-based sampling, and the survey was conducted in motor vehicle centers in the central area of Taiwan (The central areas include Nantou county, Changhwa county and Taichung city. Nantou county is defined as a rural area; Taichung city is an urban area, while Changhwa county is in-between.) All drunk drivers whose driver’s licenses are revoked for drunk driving are required to complete the road safety education courses. Questionnaires were distributed during classes to the participants in the courses from 17 August to 26 October 2020, totaling 71 days. After distributing the questionnaires, the researchers explained the questionnaires, instructed the respondents to complete the questionnaires, and then, collected all questionnaires. A total of 846 questionnaires were distributed and received, among which, 838 questionnaires were valid. The questionnaires were considered invalid if drunk drivers were riding bikes or electric bicycles when they were violating the regulation of drunk driving. Finally, 305 and 533 samples were from car (C) and motorbike (M) drivers, respectively.

The statistical analyses employed in this study included techniques of one-way and two-way exploratory analyses, EFA and AUDIT.

### 3.2. Double-Hurdle Model

Ref. [25] proposed the double-hurdle model based on a two-stage approach, deciding whether to use or not and how much to pay, that represents the decision-making of the research subjects. The popularity of the double-hurdle model in empirical work can be traced back to the works of [26,27], who are commonly associated with developing the econometric specifications of the model, as well as formally integrating it into the consumer choice theory. The double-hurdle model is specified as follows [28,29].
(1)$$y_{i1}=\omega_{i}\alpha +u_{i}$$
(2)$$y_{i2}=x_{i}\beta +v_{i}\;$$
(3)$$y_{i}=x_{i}\beta +v_{i},if,y_{i1}>0,and,y_{i2}>0$$
(4)$$y_{i}=0,\;\mathrm{otherwise}$$
where, *y_i_*_1_ is a latent endogenous variable representing the individual or household participation decision; *y_i_*_2_ is a latent endogenous variable representing the individual or household expenditure decision; *y_i_* is the observed dependent variable (expenditures); *ω_i_* is a set of individual characteristics that explain the participation decision; *x_i_* is the variable that explains the expenditure decision; and *u_i_* and *v_i_* are independent, homoscedastic, and normally distributed error terms.

To assess the impact of the repressors on the dependent variable, marginal effects can be calculated using the maximum likelihood results, as obtained from the double hurdle model. A total of three different marginal effects can be calculated based on three different definitions of the expected value of dependent variable *y_i_*. The study is most interested in the overall effect on the dependent variable, that is, the expected value of *y_i_* for values of the explanatory variables, *x*. In the Tobit model and its various generalizations, this is more commonly known as the unconditional expectation, or unconditional mean, of *y_i_* and is written as E[*y_i_* |*x*]. The unconditional expectation can be broken down into two parts, the conditional expectation of E[*y_i_* |*x, y_i_*>0], which is the expected value of *y_i_* for the values of the explanatory variables, *x*, the conditional of *y_i_* > 0, and the probability of a positive value of *y_i_* for the values of the explanatory variables, *x*, P[*y_i_* > 0 |*x*].

The decomposition of the unconditional expectation into the probability of participation and the conditional expectation is based on the work by [30] in their decomposition of the unconditional mean of the dependent variable in the Tobit model, which can be summarized by the following equation:(5)$$E\left[y_{i}|x\right]=P\left[y_{i}>0|x\right]\times E\left[y_{i}|x,y_{i}>0\right]$$

In the double-hurdle model, the probability of participation and the level of expenditures conditional on participation are [31,32]:(6)$$P\left[y_{i}>0|x\right]=\Phi \left(\omega_{i}\alpha \right)\Phi \left(\frac{x_{i}\beta }{\sigma_{i}}\right)$$
(7)$$E\left[y_{i}|y_{i}>0,x\right]=x_{i}\beta +\sigma_{i}\left(\frac{\phi \left(\frac{x_{i}\beta }{\sigma_{i}}\right)}{\Phi \left(\frac{x_{i}\beta }{\sigma_{i}}\right)}\right)$$
where, Φ(.) and ϕ(.) are the cumulative distribution function for a standard normal random variable and standard normal probability density functions, cdf and pdf, respectively. Marginal effects can be calculated by differentiating each of the above equations with respect to each explanatory variable. The following equations are given by [31,32]:(8)$$\frac{\partial P\left[y_{i}>0|x\right]}{\partial x_{j}}=\alpha_{j}\phi \left(\omega_{i}\alpha \right)\Phi \left(\frac{x_{i}\beta }{\sigma_{i}}\right)+\beta_{j}\Phi \left(\omega_{i}\alpha \right)\phi \left(\frac{x_{i}\beta }{\sigma_{i}}\right)$$
(9)$$\frac{\partial E\left[y_{i}|y_{i}>0,x\right]}{\partial x_{j}}=\beta_{j}-\beta_{j}\left(\frac{\phi \left(\frac{x_{i}\beta }{\sigma_{i}}\right)}{\Phi \left(\frac{x_{i}\beta }{\sigma_{i}}\right)}\right)\left[\frac{x_{i}\beta }{\sigma_{i}}+\left(\frac{\phi \left(\frac{x_{i}\beta }{\sigma_{i}}\right)}{\Phi \left(\frac{x_{i}\beta }{\sigma_{i}}\right)}\right)\right]$$
where, $\alpha_{j}$ and $\beta_{j}$ are the coefficients of the explanatory variable $x_{i}$ and taken from the participation and expenditure equations, respectively. Regarding the discrete explanatory variables, the estimated marginal effects represent the absolute change in the probability of a positive value, the conditional expectation, and the unconditional expectation when the value of the variable shifts from zero to one, thus, holding all the other variables constant.

The marginal effect for the unconditional level of expenditures can be derived by applying the product rule of differentiation to Equation (2):(10)$$\frac{\partial E\left[y_{i}|x\right]}{\partial x_{j}}=\frac{\partial P\left[y_{i}>0|x\right]}{\partial x_{j}}E\left[y_{i}|y_{i}>0,x\right]+\frac{\partial E\left[y_{i}|y_{i}>0,x\right]}{\partial x_{j}}P\left[y_{i}>0|x\right]$$
that is, the marginal effect of the unconditional expectation equals the marginal effect of the probability of a positive value times the conditional expectation plus the marginal effect of the conditional expectation times the probability of a positive value.

The model can be modified to allow for heteroscedasticity by specifying the variance of the errors as a function of a set of continuous variables [29,31] as follows:(11)$$\sigma_{i}=\exp\left(z_{i}h\right)$$
where, $z_{i}$ denotes the continuous variables. The exponential specification is chosen as it imposes the desirable property of the standard deviation $\sigma_{i}$ being strictly positive [31].

## 4. Data Analysis

This section provides statistics on the differences between the participants with willingness and the participants with no willingness in the two groups. The analysis focused on social-economic characteristics, awareness of the alcohol interlock and drunk driving, drinking patterns and self-health assessment characteristics, changes in the number of trips of the participants, and their willingness to use and pay for the assumed alcohol interlock scheme. The employed statistical analysis techniques include one-way and two-way exploratory analyses, EFA and AUDIT.

### 4.1. Results of Social-Economic Characteristics

As shown in Table 2, males account for about 90% of the respondents. The ages of most drunk drivers range from 36 to 55 years old. Overall, the C (car) group has a larger proportion than the M (motorbike) group. The proportion of the unmarried is larger than that of the married. In the M group, the unmarried account for the majority of the respondents with willingness. Most drunk drivers have a senior high school or vocational school degree. In the M group, most respondents with no willingness have a college degree or above. Most drunk drivers had a BAC of 0.25 mg/L and above, at the time of the incident. Overall, drunk car drivers have a higher breath alcohol concentration. Most drunk drivers have no children. Among drunk motorbike drivers, respondents with willingness account for the majority. Respondents who are raising two children have a higher proportion than those who are raising one child. The number of household cars in the C group is higher than that in the M group, and the number of household cars owned by the respondents with willingness is also higher than that of respondents with no willingness. The number of household motorbikes owned by the M group is higher than that in the C group, and the number of household motorbikes owned by the respondents with willingness is also higher than that of respondents with no willingness. Only a small number of respondents in both groups own bikes, and the number of household bikes owned by the respondents with no willingness is slightly larger than that of respondents with willingness. Most car/motorbike respondents with no willingness have an income level ranging from NTD 20,000 to 40,000; and most car/motorbike respondents with willingness have an income level of NTD 50,000 and above.

### 4.2. Awareness of Alcohol Interlock and Drunk Driving

The questionnaire of this study was developed based on [19], which discussed the effects of social factors and personality structure on drunk driving by young drivers, and by referring to the Akers’ social learning constructs scale (1 to 6 points: 1 = strongly disagree, 6 = strongly agree). This study designed questions from the aspect of personal definitions. P1 to P6 are comprised of a personal positive attitude towards the alcohol interlock. N1 to N3 are comprised of a personal negative attitude towards the alcohol interlock. V1 and V2 are comprised of a personal attitude towards risky driving. By referring to [19], the questions based on the personal definitions of the Akers social learning theory, as well as the descriptive statistics, are as listed in Table 3.

SPSS software was applied for Principal Component Analysis (PCA) for EFA on the data collected from the 838 respondents. The rotation method was Oblimin with Kaiser Normalization (KMO) with indices from 0 to 1; when the indices were 0.6 and above, they were considered as being appropriate to perform factorial analysis [19,33,34].

In this study, the KMO was 0.82 suggesting that the samples are adequate for performing factor analysis, and Bartlett’s Test of Sphericity was significant ($X^{2}$ = 5304.891; df = 55, *p* < 0.001). The result indicates that the relationship among the variables was strong, and the data were suitable to conduct EFA [35].

As listed in Table 4, three factors with eigenvalues greater than 1 were obtained through EFA. Factor 1 includes six questions, and the factor loadings ranged from 0.866 to 0.676. Factor 2 includes three questions, and the factor loadings ranged from 0.936 to 0.827. Factor 3 includes two questions, and the factor loadings ranged between 0.858 and 0.847. The reliability of the factors was assured by Cronbach’s alpha. The scores of all three factors are above 0.70, which indicate a good reliability.

### 4.3. Drinking Pattern and Self-Health Assessment Characteristics

Table 5 lists the drinking pattern and self-health assessment results based on the willingness to use alcohol interlocks. For most drunk drivers, their health condition remains unaffected after they are penalized for drunk driving. According to the AUDIT scores, the category ‘drunk car drivers’ has more high-risk drinkers than the category ‘drunk motorbike drivers’. Overall, 30% of the drunk drivers fall within the high-risk drinker group. If no efforts are made to improve their alcohol consumption patterns, they are likely to develop long-term high-risk drinking behaviors [36,37]. This can well explain the phenomenon of drunk driving [38].

### 4.4. Changes in the Number of Trips before and after Revocation

In order to understand changes in travel vehicles made by drunk car drivers after suspension or revocation of their driver’s licenses, Table 6 lists the statistics of the changes in vehicle trips before and after revocation under the condition that at least one type of vehicle was used. For respondents whose number of trips by car decreased and number of trips by motorbike increased, they transferred the number of trips from a car to a motorbike. Respondents whose number of trips by car decreased and number of trips by motorbike also decreased account for the smallest proportion, which might be because drunk drivers who were willing to reduce the number of trips by car did not reduce the number of trips by motorbike at the same time, in order to maintain their routine operations. Most respondents maintained that their number of trips by car was unchanged, that is, they still traveled by car after their driver’s licenses were suspended or revoked. This might be because they still drove their original cars upon the end of suspension or revocation. This shows that suspension and revocation of a driver’s license will only temporarily limit the use of vehicles. This might also be because drunk drivers ignore the suspension or revocation of their driver’s licenses and continue driving without a license; these drivers cause potential road safety risks. Respondents whose number of trips by motorbike remained unchanged show unapparent changes in the number of trips by car per person per week. This indicates that for respondents whose number of trips by motorbike remained unchanged, there were also no significant changes in their number of trips by car.

To understand changes in travel vehicles made by drunk motorbike drivers after suspension or revocation of their driver’s licenses, Table 7 lists the statistics on changes in vehicle trips before and after revocation under the condition that at least one type of vehicle was used. Samples whose number of trips by car increased and number of trips by motorbike decreased are similar to those in Table 6, meaning the number of trips transferred from motorbike to car is equivalent. This demonstrates that after punishment for drunk driving, the number of trips transferred from motorbike to car is equivalent. Respondents whose number of trips by car decreased and number of trips by motorbike also decreased account for the smallest proportion, which might be because drunk drivers who were willing to reduce the number of trips by car did not reduce the number of trips by motorbike at the same time, in order to maintain routine operations. This is followed by samples whose number of trips by car remained unchanged, meaning their number of trips, and number of trips per person, per week, were fewer than those of drunk drivers, whose car driver’s licenses were revoked for drunk driving, demonstrating that most drunk drivers whose motorbike driver’s licenses are revoked for drunk driving indeed use motorbikes as their main travel vehicles. Most respondents maintained an unchanged number of trips by motorbike and car per person per week, which indicates that for respondents whose number of trips by motorbike remained unchanged, there were also no significant changes in their number of trips by car.

### 4.5. Willingness to Use and Pay for the Assumed Alcohol Interlock Scheme

In the future, drunk drivers may be forced to install an alcohol interlock for one year if he/she wants to reapply for his/her driver’s license upon the end of suspension or revocation of driver’s license. The alcohol interlock price list approved by the government is provided for reference. More than half of drunk drivers are willing to buy or rent an alcohol interlock, and Table 8 lists the prices that drunk drivers are willing to pay to rent or buy an alcohol interlock. On average, drunk car drivers will accept a higher price than drunk motorbike drivers. The *p* value of the *t*-test is 0.5467 (insignificant), which indicates small differences in acceptable prices and vehicle price has an insignificant effect on drunk drivers’ willingness to pay for an alcohol interlock.

## 5. Model Estimation Results

This study applied a double-hurdle model as the theoretical basis model to discuss drunk drivers’ willingness to use and pay for alcohol interlocks. The following section uses drunk car drivers and drunk motorbike drivers as an example for demonstration purposes.

### 5.1. Drunk Car Drivers

Model calibration was executed based on the respondents with their car driver’s licenses being revoked for drunk driving. Significant variables for model calibration and calibration results are listed in Table 9 and Table 10, respectively. Regarding the significant factors, meaning Factors 1, 2, and 3 in EFA, the Least Square Regression method was applied to obtain the variables. In the double-hurdle Factor1e model, willingness to use and pay were the two most significant factors. In other words, the higher the score of drunk drivers regarding their attitude towards the alcohol interlock, the higher their willingness to use and the higher the price they are willing to pay for an alcohol interlock. AUDIT was applied to evaluate respondents’ daily drinking frequency and alcohol consumption. Respondents who were listed as high-risk drinkers according to AUDIT before and after revocation were more willing to use and pay a higher price for alcohol interlocks. Ref. [4] argued that when the risk level of drinking increases, drivers have a stronger preference for mandatory alcohol interlocks. Drivers whose health conditions have deteriorated due to alcohol consumption were more willing to use and pay more for alcohol interlocks, which might be because they anticipated improving their health conditions by participating in the alcohol interlock scheme. Drivers who participate in the alcohol interlock scheme are more likely to improve their health conditions than those who do not participate in the scheme [6]. Drunk drivers with more household cars have a better financial situation, and they are more willing to use and pay more for alcohol interlocks in exchange for the right to drive. Drunk drivers who transfer the number of trips from car to motorbike address their daily travel behaviors by using motorbikes as an alternative, thus, they experience more profoundly the penalty of losing the right to drive; therefore, they are more willing to use and pay more for alcohol interlocks. Income is a factor influencing respondents’ willingness to pay. The expense of an alcohol interlock is unaffordable for drunk drivers with a low income; therefore, they are unwilling to use and pay for alcohol interlocks. For drunk drivers whose number of trips by car have decreased, and their demands for cars also decreased, they are unwilling to use and pay for alcohol interlocks.

According to the double-hurdle marginal probability results of the drunk car drivers listed in Table 11, respondents who gave a higher score on attitude towards alcohol interlock were willing to pay NTD 2201 more than other respondents for alcohol interlocks. High-risk drinkers were willing to pay NTD 1911 more than other respondents for alcohol interlocks. Respondents who realized that their health conditions had deteriorated were willing to pay NTD 840 more for alcohol interlocks than other respondents. Respondents with more household cars were willing to pay NTD 1082 more than other respondents. Respondents who transferred the number of trips from car to motorbike were willing to pay NTD 299 more than other respondents. Low-income respondents were willing to pay NTD 2416 less than other respondents. Respondents whose number of trips by car decreased were willing to pay NTD 636 less than other respondents.

### 5.2. Drunk Motorbike Drivers

Model calibration was executed based on respondents with their motorbike drivers’ licenses being revoked for drunk driving. Significant variables for model calibration and calibration results are listed in Table 12 and Table 13, respectively. Similar to drunk car drivers, respondents who gave a higher score on attitude towards alcohol interlock were more willing to use and pay more for alcohol interlocks. Respondents who were listed as high-risk drinkers according to AUDIT before and after revocation were also more willing to use and pay more for alcohol interlocks, which might be because high-risk drinkers believe that they will become compliant by installing alcohol interlocks; however, they would not restrain their alcohol consumption modes [4]. With an increase in the number of trips, respondents had growing demands for motorbikes; therefore, they were more willing to use and pay more for alcohol interlocks in exchange for the right to drive. Drunk drivers who had an education level under college had fewer moral restrictions than those who had a college degree; thus, they were unwilling to use and pay for alcohol interlocks. Drunk drivers who raised more children coped with greater economic pressures, thus, they could not afford the expense for alcohol interlocks and were unwilling to use and pay for alcohol interlocks. Ref. [16] applied a double-hurdle model to conduct an elastic analysis of income and found that low-income families were more willing to use diesel, which is cheaper than gasoline, which demonstrates that the family economy has a significant effect on the willingness to pay.

According to the double-hurdle marginal probability results of drunk motorbike drivers listed in Table 14, respondents who gave a higher score on attitude towards alcohol interlock were willing to pay NTD 2076 more than other respondents for alcohol interlock. High-risk drinkers were willing to pay NTD 1737 more than other respondents for alcohol interlock. Respondents whose number of trips increased were willing to pay NTD 219 more than other respondents. Respondents who have an education level under college were willing to pay NTD 677 less than other respondents. Respondents who raise more children were willing to pay NTD 411 less than other respondents for alcohol interlocks.

### 5.3. Summary

The scores given by drunk drivers on the attitude towards alcohol interlocks were counted and used to generate variables by the Least Square Regression method. According to the calibration results, variables were significant in both vehicle groups of car and motorbike. This demonstrates that their personal attitude towards alcohol interlocks affects their willingness to use and pay for alcohol interlocks. According to AUDIT, high-risk drinkers will not change their willingness to use alcohol interlocks due to different vehicles, which demonstrates that when the risk level of drinking increases, drunk drivers have a stronger preference for alcohol interlocks [4]. The health factor has a significant effect on respondents’ willingness to use alcohol interlocks (Probit coefficient: 0.68). According to the research result of [6], participation in the alcohol interlock scheme for one or two years helped improve the health conditions of drunk drivers, and the odds ratio was 0.3 for one-year engagement and 8.7 for two-year engagement. Most drunk drivers whose car driver’s licenses were revoked for drunk driving addressed their daily trips by motorbike but did not reduce the number of their trips. As they experienced the inconvenience resulting from changing vehicles to a larger extent, they had stronger demands for installing alcohol interlocks in exchange for the right to drive. The Probit coefficient is 0.5, which specifies the weight affecting willingness to use. Respondents with more household cars could afford the costs and expenses of cars and were more willing to use and pay more for alcohol interlocks than those with fewer family cars. Income is a factor affecting willingness to pay. The Probit coefficient is −0.95 and the truncated coefficient is −28.83, which indicate that respondents who have an income level under NTD 10,000 per month cannot afford the expenses of alcohol interlocks. Drunk drivers whose number of trips by car decreased also reduced their demands for cars and the urgency of resuming their right to drive; therefore, they are less willing to participate in the alcohol interlock scheme and pay for alcohol interlocks. In contrast to the preceding explanatory variables, drunk motorbike drivers whose number of trips increased had growing demands for motorbikes, which in turn affected their demands for the right to drive; therefore, they were willing to participate in the alcohol interlock scheme and pay more for alcohol interlocks. Education level also affects respondents’ willingness to use and pay for alcohol interlocks. Drunk drivers who have an education level under college had less moral and legal restrictions than those who have a college degree; thus, they were less willing to participate in the alcohol interlock scheme and were willing to pay less for alcohol interlocks. Drunk drivers who raise more children coped with greater economic pressures and would measure the weight of the alcohol interlock costs and their right to drive a motorbike. As a result, in order to maintain a stable family economy, they might choose to address their daily trips by other alternatives to avoid adding the alcohol interlock costs to the family burden; therefore, they were less willing to use and pay for alcohol interlocks.

## 6. Conclusions and Suggestions

### 6.1. Conclusions

While more than 60% of drunk drivers were willing to install alcohol interlocks in exchange for the right to drive, their willingness to pay was far less than the product prices approved by the government, which explains the reason for the ineffective implementation of alcohol interlocks in Taiwan. High-risk drinkers may develop long-term risky alcohol consumption behaviors [36,37], and it is difficult for them to change their daily drinking habits to reduce their drunk driving behaviors. However, alcohol interlocks can help address road safety hazards when high-risk drinkers are willing to install alcohol interlocks to restrain their behaviors [4]. Personal attitude towards alcohol interlocks affects the respondents’ willingness to use and pay for them. By promoting road safety education courses or advertisements to increase the positive image of alcohol interlocks, the base of installed alcohol interlocks will grow.

According to the calibration results, when the risk level of drinking increases, respondents would have a stronger preference for mandatory alcohol interlocks, which might be because high-risk drinkers believe that they would be more compliant by installing alcohol interlocks. On the other hand, they would not curb their alcohol consumption modes [4]. Factors influencing the family economy, such as income and children, have a significant effect on respondents’ willingness to use and pay for alcohol interlocks. Changes in the number of daily trips also affect the respondents’ demands for the right to drive, which indirectly affects the urgency of installing alcohol interlocks.

### 6.2. Policy Implication

Suggestions are proposed based on the research results to provide a reference for future studies and policy revision.
In view of the fact that risky alcohol consumption modes are often signs of drunk driving [4], high-risk drinkers may develop long-term risky alcohol consumption behaviors[36,37], and short-term alcohol interlock usage cannot eradicate drunk driving [5]. Thus, professionals’ consultation and treatment intervention measures are considered as an important part in rectifying drunk driving issues [38,39]. Assessment diagnosis and treatments of alcohol addiction for high-risk drinkers can eradicate drunk driving, as shown in Table 3.Families with economic hardship, such as families that raise several children or have a low family income, should be granted subsidies to encourage the installation of alcohol interlocks. In 2006, the US required all drunk drivers to install alcohol interlocks, granted subsidies to low-income individuals, and required that alcohol interlock usage be monitored [40].The research results indicated that changes in the number of daily trips affected the demands for the right to drive, thus, drunk drivers would reduce the number of trips to avoid the cost of installing alcohol interlocks. Therefore, if the period of suspended or revoked driver’s licenses is shortened, drivers may be more willing to install alcohol interlocks [5].


## Figures and Tables

**Table 1 ijerph-18-11516-t001:** AUDIT statistical scale and recommendations.

Classification of Drinking Groups	Score	Recommendations and Improvements
High-risk drinking	>19 points	1. The possibility of alcohol dependence is high, and drinking habits lead to a lot of problems. 2. Stop or reduce alcohol consumption with professionals’ help. 3. A further assessment and diagnosis of alcohol addiction is needed.
Medium-risk drinking	16–19 points	1. Alcohol consumption shall be reduced because drinking habits pose serious dangers to health. 2. Professional support from doctors may be needed to reduce alcohol consumption.
Low-risk drinking	8–15 points	1. Drinking may increase the risk of compromised health. 2. Drink less and ask doctors for help.
Very low-risk drinking	1–7 points	1. To prevent cancer or other diseases, they shall consider drinking less or giving up alcohol. 2. If anyone needs to drink, male <3 units of alcohol, female <2 unit of alcohol.

**Table 2 ijerph-18-11516-t002:** Analysis of drunk car/motorbike drivers’ willingness to pay for alcohol interlock (social-economic characteristics).

**Basic Information**	**Samples with Willingness in C Group**	**Samples with No Willingness in C Group**	**Total Samplesin C Group**	**Samples with Willingness in M Group**	**Samples with No Willingness in M Group**	**Total Samples in M Group**
	**Sample (%)**	**Sample (%)**	**Sample (%)**	**Sample (%)**	**Sample (%)**	**Sample (%)**
Male	189 (90.9)	90 (92.8)	279 (91.5)	310 (87.6)	158 (88.3)	468 (87.8)
Female	19 (9.1)	7 (7.2)	26 (8.5)	44 (12.4)	21 (11.7)	65 (12.2)
18~25 years	16 (7.7)	3 (3.1)	19 (6.2)	38 (10.7)	16 (8.9)	54 (10.1)
26~35 years	29 (13.9)	22 (22.7)	51 (16.7)	74 (20.9)	21 (11.7)	95 (17.8)
36~45 years	71 (34.1)	28 (28.9)	99 (32.5)	96 (27.1)	49 (27.4)	145 (27.2)
46~55 years	60 (28.8)	26 (26.8)	86 (28.2)	96 (27.1)	58 (32.4)	154 (28.9)
>55 years	32 (15.4)	18 (18.6)	50 (16.4)	50 (14.1%)	35 (19.6)	85 (15.9)
Married	98 (47.1)	43 (44.3)	141 (46.2)	140 (39.5)	79 (44.1)	219 (41.1)
Unmarried	110 (52.9)	54 (55.7)	164 (53.8)	214 (60.5)	100 (55.9)	314 (58.9)
Elementary	4 (1.9)	3 (3.1)	7 (2.3)	5 (1.4)	7 (3.9)	12 (2.3)
Junior high	44 (21.2)	27 (27.8)	71 (23.3)	62 (17.5)	33 (18.4)	95 (17.8)
Senior high	91 (43.8)	39 (40.2)	130 (42.6)	167 (47.2)	93 (52.0)	260 (48.8)
Junior college	18 (8.7)	7 (7.2)	25 (8.2)	25 (7.1)	18 (10.1)	43 (8.1)
University	51 (24.5)	21 (21.6)	72 (23.6)	95 (26.8%)	28 (15.6%)	123 (23.1)
Minor violation	24 (36.9)	71 (29.6)	95 (31.1)	116 (32.8)	63 (35.2)	179 (33.6)
Serious violation	38 (58.5)	156 (65.0)	194 (63.6)	216 (61.0)	108 (60.3)	324 (60.8)
Refusal violation	3 (4.6)	13 (5.4)	16 (5.2)	22 (6.2)	8 (4.5)	30 (5.6)
Children = 0	82 (39.4)	40 (41.2)	122 (40.0)	194 (54.8)	78 (43.6)	272 (51.0)
Children = 1	36 (17.3)	16 (16.5)	52 (17.0)	45 (12.7)	26 (14.5)	71 (13.3)
Children = 2	57 (27.4)	25 (25.8)	82 (26.9)	74 (20.9)	51 (28.5)	125 (23.5)
Children > 2	33 (15.9)	16 (16.5)	49 (16.1)	41 (11.6)	24 (13.4)	65 (12.2)
Total	208 (100)	97 (100)	305 (100)	99 (100)	434 (100)	533 (100)
**Basic Information**	**Samples with Willingness in C Group**	**Samples with No Willingness in C Group**	**Total Samplesin C Group**	**Samples with Willingness in M Group**	**Samples with No Willingness in M Group**	**Total Samplesin M Group**
	**Sample (%)**	**Sample (%)**	**Sample (%)**	**Sample (%)**	**Sample (%)**	**Sample (%)**
Car = 0	3 (4.6)	25 (10.4)	28 (9.2)	136 (38.4)	72 (40.2)	208 (39.0)
Car = 1	41 (63.1)	136 (56.7)	177 (58.0)	148 (41.8)	70 (39.1)	218 (40.9)
Car = 2	13 (20.0)	54 (22.5)	67 (22.0)	48 (13.6)	29 (16.2)	77 (14.4)
Car > 2	8 (12.3)	25 (10.4)	33 (10.8)	22 (6.2)	8 (4.5)	30 (5.6)
Average	1.41/person	1.20/person	1.34/person	0.88/person	0.85/person	0.87/person
Motorbike = 0	14 (21.5)	47 (19.6)	61 (20.0)	18 (5.1)	4 (2.2%)	22 (4.1)
Motorbike = 1	24 (36.9)	97 (40.4)	121 (39.7)	151 (42.7)	96 (53.6)	247 (46.3)
Motorbike = 2	14 (21.5)	51 (21.3)	65 (21.3)	94 (26.6)	47 (26.3)	141 (26.5)
Motorbike = 3	7 (10.8)	28 (11.7)	35 (11.5)	53 (15.0)	20 (11.2)	73 (13.7)
Motorbike > 3	6 (9.2)	17 (7.1)	23 (7.5)	38 (10.7)	12 (6.7)	50 (9.4)
Average	1.53/person	1.34/person	1.47/person	1.84/person	1.66/person	1.78/person
Bike = 0	53 (81.5)	189 (78.8)	242 (79.3)	82 (82.8)	282 (79.7)	142 (79.3)
Bike = 1	6 (9.2)	25 (10.4)	31 (10.2)	11 (11.1)	38 (10.7)	20 (11.2)
Bike > 1	6 (9.2)	26 (10.8)	32 (10.5)	6 (6.1)	34 (9.6)	17 (9.5)
Average	0.40/person	0.25/person	0.35/person	0.30/person	0.30/person	0.30/person
Income NTD < 20,000	25 (12.0)	30 (30.9)	25 (12.0)	71 (20.1)	53 (29.6)	124 (23.3)
20,000~30,000	42 (20.2)	20 (20.6)	42 (20.2)	89 (25.1)	37 (20.7)	126 (23.6)
30,000~40,000	44 (21.2)	20 (20.6)	44 (21.2)	81 (22.9)	46 (25.7)	127 (23.8)
40,000~50,000	32 (15.4)	10 (10.3)	32 (15.4)	59 (16.7)	20 (11.2)	79 (14.8)
50,000~60,000	21 (10.1)	5 (5.2)	21 (10.1)	20 (5.6)	11 (6.1)	31 (5.8)
60,000~80,000	20 (9.6)	5 (5.2)	20 (9.6)	12 (3.4)	6 (3.4)	18 (3.4)
>80,000	24 (11.5)	7 (7.2)	24 (11.5)	22 (6.2)	6 (3.4)	28 (5.3)
Total	208 (100)	97 (100)	305 (100)	354 (100)	179 (100)	533 (100)

**Table 3 ijerph-18-11516-t003:** Descriptive statistics of questions on attitude towards alcohol interlock and reference question IDs.

Reference Question ID	Question	Code	Mean	Std. Deviation
1	I know the policy that I must install an alcohol interlock if I want to reapply for my driver’s license after it is revoked.	P1	3.337	1.7210
2	Under the current policy, I am willing to install an alcohol interlock.	P2	2.951	1.6253
3	I think that my drinking problem will be improved if an alcohol interlock is installed.	P3	3.037	1.6756
4	I think my friends and families will support me in installing an alcohol interlock.	P4	3.019	1.6776
5	I know very well about the penalty provisions if an alcohol interlock is not installed.	P5	3.407	1.6743
6	I believe that alcohol interlocks will make me curb my behavior.	P6	3.295	1.6707
7	I think alcohol interlocks are too expensive.	N1	4.669	1.6225
8	I think an alcohol interlock will violate my privacy.	N2	4.055	1.7191
9	I think an alcohol interlock will violate my autonomy.	N3	4.013	1.7292
10	I often drive without complying with traffic rules.	V1	2.243	1.4642
11	I think it is safe to drive after drinking.	V2	2.038	1.4171

**Table 4 ijerph-18-11516-t004:** Exploratory factor analysis.

Rotated Component Matrix ^a^			
Item Code	Component		
	Factor 1	Factor 2	Factor 3
P1	0.866	−0.024	0.060
P2	0.862	−0.055	0.077
P3	0.844	−0.033	0.171
P4	0.800	0.017	0.157
P5	0.785	−0.011	0.037
P6	0.676	0.129	0.095
O1	−0.074	0.936	0.136
O2	−0.099	0.925	0.136
O3	0.183	0.827	−0.081
B1	0.110	0.121	0.858
B2	0.198	0.024	0.847
Extraction Method: Principal Component Analysis Rotation Method: Varimax with Kaiser Normalization			

^a^ Rotation converged in 4 iterations.

**Table 5 ijerph-18-11516-t005:** Drinking pattern and self-health assessment statistics.

Basic Information	Samples with Willingness in C Group	Samples with No Willingness in C Group	Total Samplesin C Group	Samples with Willingness in M Group	Samples with No Willingness in M Group	Total Samplesin M Group
	Sample (%)	Sample (%)	Sample (%)	Sample (%)	Sample (%)	Sample (%)
Health Improvement	17 (8.2)	8 (8.2)	25 (8.2)	20 (5.6)	12 (6.7)	32 (6.0)
Health Deterioration	24 (11.5)	5 (5.2)	29 (9.5)	25 (7.1)	12 (6.7)	37 (6.9)
Health Unchanged	169 (81.3)	82 (84.5)	251 (82.3)	323 (91.2)	141 (78.8)	464 (87.1)
AUDIT > 19	62 (29.8)	31 (32.0)	93 (30.5)	104 (29.4)	44 (24.6)	148 (27.8)
AUDIT = 16~19	13 (6.3)	2 (2.1)	15 (4.9)	50 (14.1)	14 (7.8)	64 (12.0)
AUDIT = 8~15	74 (35.6)	21 (21.6)	95 (31.1)	112 (31.6)	49 (27.4)	161 (30.2)
AUDIT = 1~7	20 (9.6)	16 (16.5)	36 (11.8)	47 (13.3)	27 (15.1)	74 (13.9)
AUDIT = 0	39 (18.8)	27 (27.8)	66 (21.6)	55 (15.5)	31 (17.3)	86 (16.1)
Total	208 (100)	97 (100)	305 (100)	99 (100)	434 (100)	533 (100)

**Table 6 ijerph-18-11516-t006:** Analysis of vehicle changes before and after prohibition, and the willingness and unwillingness of car drivers to use alcohol interlock.

Changes in the Number of Trips	Vehicles	Willing			Unwilling		
		Samples	Before	After	Samples	Before	After
(Cars, motorbikes) (Decrease, increase)	cars↓	38	693 (2.61) *	123 (0.46)	11	817 (10.61)	56 (0.73)
	motorbikes↑		201 (0.76)	756 (2.84)		16 (0.21)	740 (9.61)
(Cars, motorbikes) (Decrease, increase)	cars↓	12	336 (4.00)	39 (0.46)	5	46 (1.31)	16 (0.46)
	motorbikes↓		316 (3.76)	97 (1.15)		155 (0.82)	116 (3.31)
(Cars, motorbikes) (Unchange, decrease)	cars–	63	1406 (3.19)		27	532 (2.81)	
	motorbikes↓		191 (0.43)	191 (0.43)		155 (0.82)	125 (0.66)
(Cars, motorbikes) (Decrease, unchange)	cars↓	31	197 (0.69)	169 (0.59)	18	162 (1.29)	161 (1.28)
	motorbikes–		424 (1.48)			323 (2.56)	

* The figures in brackets are the average number of trips per person per week.

**Table 7 ijerph-18-11516-t007:** Analysis of vehicle changes before and after prohibition, and the willingness and unwillingness of motorbike drivers to use alcohol interlock.

Changes in theNumber of Trips	Vehicles	Willing			Unwilling		
		Samples	Before	After	Samples	Before	After
(Cars, motorbikes) (Decrease, increase)	cars↑	37	133 (0.51) *	703 (2.71)	15	30 (0.29)	283 (2.70)
	motorbikes↓		748 (2.89)	60 (0.23)		313 (2.98)	9 (0.009)
(Cars, motorbikes) (Decrease, decrease)	cars↓	14	152 (1.55)	53 (0.54)	11	135 (1.75)	42 (0.55)
	motorbikes↓		236 (2.41)	64 (0.65)		170 (2.21)	15 (0.19)
(Cars, motorbikes) (Unchanged, decrease)	cars–	59	550 (1.33)		24	553 (3.29)	
	motorbikes↓		622 (1.51)	502 (1.22)		319 (1.90)	269 (1.60)
(Cars, motorbikes) (Decrease, unchanged)	cars↓	132	281 (0.30)	259 (0.28)	61	329 (0.77)	327 (0.77)
	motorbikes–		2464 (2.67)			974 (2.28)	

* The figures in brackets are the average number of trips per person per week.

**Table 8 ijerph-18-11516-t008:** Statistics on willingness to use and pay for the assumed alcohol interlock.

Statistics Variables	Car			Motorbike		
	Buy	Rent	Unwilling	Buy	Rent	Unwilling
Samples (%)	208 (68.2%)	205 (67.2%)	92 (30.2%)	354 (66.4%)	353 (66.2%)	164 (30.8%)
Average price	NTD12,803	NTD 1005/month		NTD12,105	NTD 956/month	
Maximum value	NTD110,000	NTD 10,000/month		NTD100,000	NTD 10,000/month	
Minimum value	NTD500	NTD 30/month		NTD 200	NTD 50/month	
Mode (%)	NTD10,000 (37.5%)	NTD 500/month (40.0%)		NTD10,000 (37%)	NTD 500/month (41.6%)	

**Table 9 ijerph-18-11516-t009:** Variables used in the double-hurdle model of car drivers’ willingness to use alcohol interlocks.

Variables	Interpreted Meaning	Min	Max	Mean	Std.
Factor 1	Factor 1 variable obtained by using the Least Square Regression method	−1.58	2.08	−0.12	1.01
AUDIT	High-risk drinkers before and after revocation	0	1	0.50	0.48
H1	Deterioration of health conditions	0	1	0.10	0.29
HC	Number of cars owned	0	2	1.24	0.60
N1	Number trips transferred from car to motorbike	0	1	0.16	0.37
IC1	Income level under NTD 10,000/month	0	1	0.07	0.25
WC1	Decrease in number of trips by car	0	1	0.11	0.32

**Table 10 ijerph-18-11516-t010:** Double-hurdle model calibration table for drunk are drivers.

Variables	Probit	Truncated
	Coef. (t-Value)	
Constant	−0.06(−0.19)	−118.34(−2.48 **)
Factor 1	0.26(3.27 ***)	26.87(2.80 ***)
AUDIT	0.29(1.77 *)	23.26(1.46)
H1	0.68(2.21 **)	9.67(0.44)
HC	0.25(1.93 *)	13.08(1.06)
N1	0.50(1.95 *)	3.19(0.14)
IC1	−0.95(−3.88 ***)	−28.83(−0.95)
WC1	−0.39(−1.88 *)	−7.46(−0.39)
Sigma	-	35.29(6.06 ***)
Log likelihood function	−167.69	−739.46
Pseudo R-squared	0.114	-
WTP	-	NTD 10,808
Number of samples	305	

*** α = 1%; ** α = 5%; * α = 10%.

**Table 11 ijerph-18-11516-t011:** Double-hurdle marginal probability results of drunk car drivers.

Variables	Willingness to Use Status *	Willingness to Pay Status	Price **
Factor 1	0.01%	0.81%	2201
AUDIT	0.01%	0.70%	1911
H1	0.02%	0.29%	840
HC	0.01%	0.39%	1082
N1	0.02%	0.10%	299
IC1	−0.03%	−0.86%	−2416
WC1	−0.01%	−0.22%	−636

* Use Equation (8); ** Use Equation (9).

**Table 12 ijerph-18-11516-t012:** Variables used in the double-hurdle model of motorbike drivers’ willingness to use alcohol interlocks.

Variables	Interpreted Meaning	Min	Max	Mean	Std.
Factor 1	Factor 1 variable obtained by using the Least Square Regression method	−1.58	2.08	0.07	0.99
AUDIT	High-risk drinkers before and after revocation	0	1	0.53	0.50
TW2	Increase in number of trips by car	0	1	0.06	0.24
ED1-4	Education level under college	0	1	0.77	0.42
CM	Number of children	0	6	0.99	1.17

**Table 13 ijerph-18-11516-t013:** Double-hurdle model calibration table for drunk motorbike drivers.

Variable	Probit	Truncated
	Coef. (t-Value)	
Constant	0.74(4.98 ***)	−76.87(−5.53 ***)
Factor 1	0.23(3.82 ***)	25.30(6.14 ***)
AUDIT	0.22(1.91 *)	21.14(2.71 ***)
TW2	0.45(1.67 *)	2.24(0.14)
ED1-4	−0.34(−2.26 **)	−7.98(−0.90)
CM	−0.12(−2.32 **)	−4.94(−1.51)
Sigma	-	27.81(15.26 ***)
Log likelihood function	−312.26	−1662.77
Pseudo R-squared	0.053	-
WTP	-	8529
Number of samples	533	

*** α = 1%; ** α = 5%; * α = 10%.

**Table 14 ijerph-18-11516-t014:** Double-hurdle marginal probability results of drunk motorbike drivers.

Variable	Willingness to Use Status *	Willingness to Pay Status	Price **
Factor 1	0.01%	0.98%	2076
AUDIT	0.01%	0.82%	1737
TW2	0.02%	0.09%	219
ED1-4	−0.01%	−0.31%	−677
CM	0.00%	−0.19%	−411

* Use Equation (8); ** Use Equation (9).

## Data Availability

Data will be provided upon request.

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
