# Peer review of "Car/Motorbike Drivers’ Willingness to Use and to Pay for Alcohol Interlock in Taiwan"

_ijerph, 2021, doi:10.3390/ijerph182111516_

Round 1

Reviewer 1 Report

The paper deals with a very interesting subject, examining factors influencing willingness to pay for and use alcolock. It uses an appropriate methodological approach and produces promising and novel research results. Overall, the paper is very well written, however, tables are badly formatted to the point that the reviewer cannot clearly perceive much of the relevant information. I would ask for the authors to provide a clearer version so that I can read the paper properly and provide a proper review. There are additional small issues in the present form of the paper that can be addressed at this stage.

  • On lines 14-16 of the abstract, it is mentioned that “..finds that the significant variables that influence willingness to participate and to pay are recognition of alcohol interlocks, the number of cars owned, university degree or below, and low incomes.” Some of these are not variables themselves, but categorical values that are significant from specific variables. Please rewrite the phrase to be more accurate. The subsequent explanations are clearer, for instance.
  • The authors are suggested to define what ‘A1 class’ accidents are, as mentioned on line 32 of page 1, because the majority of readers are unfamiliar with that term.
  • The description of legislation/fines on page 2, lines 45-68 of the introduction, is unnecessary and redundant. A small overview is more than sufficient.
  • The literature review could benefit from some pertinent studies which examine several topics similar to those considered by the authors, especially on both alcohol and alcolock behavior and their impacts on road safety. Indicative references are:

Beck, K. H., Scherer, M., Romano, E., Taylor, E., & Voas, R. (2020). Driver experiences with the alcohol ignition interlock: Comparing successful and poor performers. Traffic injury prevention, 21(7), 413-418.

Manan, M. M. A., Ho, J. S., Arif, S. T. M. S. T., Ghani, M. R. A., & Várhelyi, A. (2017). Factors associated with motorcyclists’ speed behaviour on Malaysian roads. Transportation research part F: traffic psychology and behaviour, 50, 109-127.

Ziakopoulos, A., Nikolaou, D., & Yannis, G. (2021). Correlations of multiple rider behaviors with self-reported attitudes, perspectives on traffic rule strictness and social desirability. Transportation Research Part F: Traffic Psychology and Behaviour, 80, 313-327.

And also recent studies regarding factor analysis:

Sullman, M. J., Stephens, A. N., & Taylor, J. E. (2019). Dimensions of aberrant driving behaviour and their relation to crash involvement for drivers in New Zealand. Transportation research part F: traffic psychology and behaviour, 66, 111-121.

Ziakopoulos, A., Theofilatos, A., Laiou, A., Michelaraki, E., Yannis, G., & Rosenbloom, T. (2021). Examining the relationship between impaired driving and past crash involvement in Europe: Insights from the ESRA study. International journal of injury control and safety promotion, 1-11.

Ojsteršek, T. C., & Topolšek, D. (2019). Influence of drivers’ visual and cognitive attention on their perception of changes in the traffic environment. European transport research review, 11(1), 1-9.

  • Apart from the mentioned choice-based sampling, the authors distributed the questionnaires to motor vehicle stations in central areas. Is there a recruitment bias by excluding highly rural areas, where people are known to consume alcohol heavily? The authors are asked to elaborate.
  • The findings mentioned at the start of page 42 are interesting. The authors are asked to elaborate on top of the link to Bishop et al. (2017) (also mentioned on lines 723 & 761). Are there any indicators that the driver sample would want more enforcement/discipline despite personal behavior? Can results be generalized?
  • The reviewer is skeptical of the changes mentioned on Table 18 and the respective commentary. Is a finding of ±0.02% Willingness-to-Pay important to mention or consider for any practical applications? Please discuss or remove this section (or at the very least shorten the explanations) otherwise.
  • Obviously, the paper format seems to have gone wrong regarding the tables, at least on the pdf version of the reviewer. As such, they are impossible to review. Please provide a corrected version.

    Thank you and best wishes!

Author Response

The paper deals with a very interesting subject, examining factors influencing willingness to pay for and use alcolock. It uses an appropriate methodological approach and produces promising and novel research results. Overall, the paper is very well written, however, tables are badly formatted to the point that the reviewer cannot clearly perceive much of the relevant information. I would ask for the authors to provide a clearer version so that I can read the paper properly and provide a proper review.

R: Sorry for the error. The problem has been fixed thanks to the Journal’s editing team.

 There are additional small issues in the present form of the paper that can be addressed at this stage.

On lines 14-16 of the abstract, it is mentioned that “..finds that the significant variables that influence willingness to participate and to pay are recognition of alcohol interlocks, the number of cars owned, university degree or below, and low incomes.” Some of these are not variables themselves, but categorical values that are significant from specific variables. Please rewrite the phrase to be more accurate.

R: We have rephrased the related materials as follows. “This study respectively estimates the variables affecting drunk car/motorbike drivers by the Double-Hurdle mode. The results indicate two variables are significant in both car/motorbike models. The first one is the respondents who are classified by AUDIT as high-risk drinkers before and after drunk driving prohibition. Those drivers are more willing to use alcohol interlocks and to pay higher prices. The second one is the respondents with declined health self-assessments. Those drivers are also more willing to use alcohol interlocks and pay higher prices, probably because they are aware of their declining health and would like to improve their health by participating in the alcohol interlock program.”

The subsequent explanations are clearer, for instance,The authors are suggested to define what ‘A1 class’ accidents are, as mentioned on line 32 of page 1, because the majority of readers are unfamiliar with that term.

R: Please refer to footnote 1 for the definition of A1 class. “A1 class means someone is killed instantly or dies within 24 hours of when the accident occurred regardless of hospitalization or not.”

The description of legislation/fines on page 2, lines 45-68 of the introduction, is unnecessary and redundant. A small overview is more than sufficient.

R: Thank you for the comment. We have shortened the description as follows. “The Ministry of Transportation and Communications implemented a new policy on drunk driving in March 2020: anyone with a breath alcohol concentration more than 0.15mg/L or a blood alcohol concentration more than 0.03% is considered a drunk driver, and fines vary according to the vehicle, ranging from NTD[1] 30,000 to NTD 120,000 for cars, NTD 30,000 to NTD 90,000 for motorbikes. Those who drive drunk more than three times shall receive treatment for alcohol addiction before taking the test to get new licenses for cars and motorbikes, then the supervision authority will issue one-year restricted driver’s licenses. Drivers with restricted driver’s licenses are only allowed to use restricted vehicles equipped with alcohol interlocks (Traffic Safety Committee, the Ministry of Transportation). In Taiwan, while a large number of applicants participated in the driver’s license examination from March 2020 to December 2020 after their driver’s licenses were suspended, only a few have installed qualified alcohol interlocks and registered with the supervision authority, which indicates the insignificant effect of the alcohol interlock policy.”

The literature review could benefit from some pertinent studies which examine several topics similar to those considered by the authors, especially on both alcohol and alcolock behavior and their impacts on road safety. Indicative references are:

Beck, K. H., Scherer, M., Romano, E., Taylor, E., & Voas, R. (2020). Driver experiences with the alcohol ignition interlock: Comparing successful and poor performers. Traffic injury prevention, 21(7), 413-418.

R: Added. “Beck et al. (2020) indicated that drunk drivers with a more serious drinking pattern and who have a higher risk for recidivating may more intend to recognize the potential benefits of the interlock as a DUIA (Driving under the influence of alcohol) preventive countermeasure.” Please refer to page 6.

Manan, M. M. A., Ho, J. S., Arif, S. T. M. S. T., Ghani, M. R. A., & Várhelyi, A. (2017). Factors associated with motorcyclists’ speed behaviour on Malaysian roads. Transportation research part F: traffic psychology and behaviour, 50, 109-127.

R: We have added in Section 1. “(4) to explore drunk car drivers and drunk motorbike drivers, respectively. The inclusion of motorbike drivers has two reasons. One is traffic composition in Taiwan involves a higher share of motorcycles in the market. The other one is the dangerous behavior and vulnerable characteristics of motorbike drivers under the influence of Alcohol (Mana, 2017; Ziakopoulos et al., 2021)”.

Ziakopoulos, A., Nikolaou, D., & Yannis, G. (2021). Correlations of multiple rider behaviors with self-reported attitudes, perspectives on traffic rule strictness and social desirability. Transportation Research Part F: Traffic Psychology and Behaviour, 80, 313-327.

R: We have added in Section 1. “(4) to explore drunk car drivers and drunk motorbike drivers, respectively. The inclusion of motorbike drivers has two reasons. One is traffic composition in Taiwan involves a higher share of motorcycles in the market. The other one is the dangerous behavior and vulnerable characteristics of motorbike drivers under the influence of Alcohol (Mana, 2017; Ziakopoulos et al., 2021)”

And also recent studies regarding factor analysis:

Sullman, M. J., Stephens, A. N., & Taylor, J. E. (2019). Dimensions of aberrant driving behaviour and their relation to crash involvement for drivers in New Zealand. Transportation research part F: traffic psychology and behaviour, 66, 111-121.

R: Added. “The paper done by Sullman et al. (2019) used factor analysis to identify the most suitable factor structure for drivers in New Zealand and to understand the associations between factor outcomes and crash involvement.” 

Ziakopoulos, A., Theofilatos, A., Laiou, A., Michelaraki, E., Yannis, G., & Rosenbloom, T. (2021). Examining the relationship between impaired driving and past crash involvement in Europe: Insights from the ESRA study. International journal of injury control and safety promotion, 1-11.

R: Added. “Ziakopoulos et al. (2021) applied Principal Component Analysis (PCA) to consolidate relative questions in numeric factor quantities, and then logistic regression was implemented on the calculated component scores to investigate the effects of each factor on past crash involvement of car drivers. “

Ojsteršek, T. C., & Topolšek, D. (2019). Influence of drivers’ visual and cognitive attention on their perception of changes in the traffic environment. European transport research review, 11(1), 1-9.

R: Added. “Ojsteršek et al. (2019) used driver self evaluation data to define which elements cause visual and cognitive distraction. Exploratory factor analysis (EFA), Confirmatory factor analysis (CFA), and Structural Equation Modelling (SEM) were implemented. Results showed that the impacts of different factors on drivers’ perception of crucial changes in the traffic environment were varies.”

Apart from the mentioned choice-based sampling, the authors distributed the questionnaires to motor vehicle stations in central areas. Is there a recruitment bias by excluding highly rural areas, where people are known to consume alcohol heavily? The authors are asked to elaborate.

R: Sorry for the confusion. The central areas include Nantou county, Changhwa county and Taichung city. Nantou county is defined as a rural area, Taichung city is an urban area, while Changhwa county is in between. The effect of city was tried to capture by estimating the city variable. Nevertheless, it was not statistically significant (significant level is 10%) to be included in the models.

The findings mentioned at the start of page 42 are interesting. The authors are asked to elaborate on top of the link to Bishop et al. (2017) (also mentioned on lines 723 & 761). Are there any indicators that the driver sample would want more enforcement/discipline despite personal behavior? Can results be generalized?

R: Please refer to the first point of Section 6.2. “In view of the fact that risky alcohol consumption modes are often signs of drunk driving (Bishop, 2017), high-risk drinkers may develop long-term risky alcohol consumption behaviors (DiClemente et al., 1999; Miller & Tonigan, 1996), and short-term alcohol interlock usage cannot eradicate drunk driving (Tracey et al., 2016), thus,  professionals’ consultation and treatment intervention measures are considered an important part of rectifying drunk driving issues (Filtness et al., 2015, Fitzharris et al., 2015). Assessment diagnosis and treatment of alcohol addiction for high-risk drinkers can eradicate drunk driving (Please refer to Table 3).”

The reviewer is skeptical of the changes mentioned on Table 18 and the respective commentary. Is a finding of ±0.02% Willingness-to-Pay important to mention or consider for any practical applications? Please discuss or remove this section (or at the very least shorten the explanations) otherwise.

R: Thank you for the comment. We have rewritten the paragraph as follows. “According to the double-hurdle marginal probability results of motorbike drunk drivers, as listed in Table 18, participants who gave a higher score on attitude towards alcohol interlock are willing to pay NTD 2076 more than other participants for alcohol interlock. High-risk drinkers are willing to pay NTD 1737 more than other participants for alcohol interlock. Participants whose number of trips increased are willing to pay NTD 219 more than other participants. Participants who have an education level under college are willing to pay NTD 677 less than other participants. Participants who raise more children are willing to pay NTD 411 less than other participants for alcohol interlock.

Obviously, the paper format seems to have gone wrong regarding the tables, at least on the pdf version of the reviewer. As such, they are impossible to review. Please provide a corrected version.

R: Thank you for your patient.

Thank you and best wishes!

Reviewer 2 Report

The manuscript novelty and the specific objective are not clearly stated along the text.

Right in the abstract, there is no scientific structure related to context, objective, methodology... Although some recommendations are mentioned, the methodology and result discussion are very hard to follow.

The Introduction section is fairly well written, but the overall structure is not adequate and needs major revision to ease the reading. Major flaws can be pointed, due to several formatting errors along the manuscript (the number of pages of the manuscript is unrealistic, Table 1 format is awkward, various pages with white huge spaces, just with a word, acronyms without spelling - e.g., BAC, ...), or the written style that sometimes does not seem scientific. For instance, it is very strange that it appears on page 13 of the pdf some link associated to a question "2. How many drinks containing alcohol do you have on a typical day when you are drinking?", but it goes for the paper from 2019 "Alcohol and traffic accidents in Japan" with very similar topic/questions and no reference to this is included in the manuscript in the literature review - this is not the correct attitude.

In mathematical terms, I also think the authors should revise all the equations and the scientific language.

I strongly suggest the authors to carefully revise and prepare a quality manuscript.

Author Response

The manuscript novelty and the specific objective are not clearly stated along the text.

Right in the abstract, there is no scientific structure related to context, objective, methodology... Although some recommendations are mentioned, the methodology and result discussion are very hard to follow.

R: Thanks for your comments. We have revised accordingly. Please refer to Abstract.

“The objective of this study is to explore the important factors affecting drunk car/motorbike drivers’ willingness to use and pay for alcohol interlocks. Data was obtained through the survey conducted in center Taiwan.”

“The method used in this study applies the Double-Hurdle model which can estimate the variables affecting drunk car/motorbike drivers.”

“The results indicate two variables are significant in both car/motorbike models. The first one is the respondents who are classified by AUDIT as high-risk drinkers before and after drunk driving prohibition. Those drivers are more willing to use alcohol interlocks and to pay higher prices. The second one is the respondents with declined health self-assessments.”

“This study suggests that the following measures or policies should be taken to increase the alcohol interlock installation rate….”

The Introduction section is fairly well written, but the overall structure is not adequate and needs major revision to ease the reading. Major flaws can be pointed, due to several formatting errors along the manuscript (the number of pages of the manuscript is unrealistic, Table 1 format is awkward, various pages with white huge spaces, just with a word, acronyms without spelling - e.g., BAC, ...), or the written style that sometimes does not seem scientific. For instance, it is very strange that it appears on page 13 of the pdf some link associated to a question "2. How many drinks containing alcohol do you have on a typical day when you are drinking?", but it goes for the paper from 2019 "Alcohol and traffic accidents in Japan" with very similar topic/questions and no reference to this is included in the manuscript in the literature review - this is not the correct attitude.

R: Sorry for the error. The problem has been fixed thanks to the Journal for editing the format. Also, the acronym BAC stands for breath alcohol concentration and NEO stands for Neuroticism-Extraversion-Openness. Please refer to pages 2 and 9. The links in Table 2 are not necessary, therefore were removed, sorry for the mistake.

In mathematical terms, I also think the authors should revise all the equations and the scientific language.

R: Sorry for the error. We have revised all the equations and used the proper scientific language.

I strongly suggest the authors to carefully revise and prepare a quality manuscript.

R: Thank the reviewer for his/her suggestions.

Round 2

Reviewer 1 Report

Major revision completed

The authors have taken into account all of the previous comments, and tackled them efficiently and satisfactorily.

Minor observations now include:

  1. The authors should include the following explanation that they provided in response to a reviewer comment within the manuscript:
    “The central areas include Nantou county, Changhwa county and Taichung city. Nantou county is defined as a rural area, Taichung city is an urban area, while Changhwa county is in between.”
  2. Please provide a minor explanation on Tables 15 & 18 on how the marginal probability was extracted from the model and calculated after the model was trained, as it is not very clear at present.
  3. The fixed tables are fine.

Apart from these, the paper is ready for publication and should therefore be accepted.

Author Response

The authors have taken into account all of the previous comments, and tackled them efficiently and satisfactorily.

R: Thank you for your positive feedback.

Minor observations now include:

  1. The authors should include the following explanation that they provided in response to a reviewer comment within the manuscript:
    “The central areas include Nantou county, Changhwa county and Taichung city. Nantou county is defined as a rural area, Taichung city is an urban area, while Changhwa county is in between.”

         R: Done.

  1. Please provide a minor explanation on Tables 15 & 18 on how the marginal probability was extracted from the model and calculated after the model was trained, as it is not very clear at present.

          R: Added. Please refer to Tables 15 and 18.

  1. The fixed tables are fine.

Apart from these, the paper is ready for publication and should therefore be accepted.

R: Appreciate for your support.

Reviewer 2 Report

The authors have addressed most of my concerns and suggestions. The manuscript readability has significantly improved.

It seems that it still presents some formatting issues in terms of fonts used, e.g., in footnotes, in equation numeration,...

Overall, the manuscript has good quality and can be considered for possible publication in the Journal.

Author Response

The authors have addressed most of my concerns and suggestions. The manuscript readability has significantly improved.

R: Thank you.

It seems that it still presents some formatting issues in terms of fonts used, e.g., in footnotes, in equation numeration,...

R: Fixed.

Overall, the manuscript has good quality and can be considered for possible publication in the Journal.

R: Appreciate.